# The Complex FtBBX22 and FtHY5 Positively Regulates Light-Induced Anthocyanin Accumulation by Activating *FtMYB42* in Tartary Buckwheat Sprouts

**DOI:** 10.3390/ijms25158376

**Published:** 2024-07-31

**Authors:** Jiao Deng, Lan Zhang, Lijuan Wang, Jiali Zhao, Chaojie Yang, Hongyou Li, Juan Huang, Taoxiong Shi, Liwei Zhu, Rebecca Njeri Damaris, Qingfu Chen

**Affiliations:** 1Research Center of Buckwheat Industry Technology, School of Life Sciences, Guizhou Normal University, Guiyang 550001, China; 201507012@gznu.edu.cn (J.D.); zhangl8720@163.com (L.Z.); ljw58273@163.com (L.W.); 15685965728m@sina.cn (C.Y.); lihongyouluod@163.com (H.L.); huang200669@163.com (J.H.); shitaoxiong@gznu.edu.cn (T.S.); liweib0401001@163.com (L.Z.); 2School of Life Sciences, Sichuan Agricultural University, Ya’an 625099, China; zhaojl1996@163.com; 3Department of Biological Sciences, Pwani University, Kilifi 195-80108, Kenya

**Keywords:** Tartary buckwheat, anthocyanin biosynthesis, light induction, *FtBBX22*, *FtHY5*, *FtMYB42*

## Abstract

Anthocyanin is one important nutrition composition in Tartary buckwheat (*Fagopyrum tataricum*) sprouts, a component missing in its seeds. Although anthocyanin biosynthesis requires light, the mechanism of light-induced anthocyanin accumulation in Tartary buckwheat is unclear. Here, comparative transcriptome analysis of Tartary buckwheat sprouts under light and dark treatments and biochemical approaches were performed to identify the roles of one B-box protein BBX22 and ELONGATED HYPOCOTYL 5 (HY5). The overexpression assay showed that FtHY5 and FtBBX22 could both promote anthocyanin synthesis in red-flower tobacco. Additionally, FtBBX22 associated with FtHY5 to form a complex that activates the transcription of MYB transcription factor genes *FtMYB42* and *FtDFR*, leading to anthocyanin accumulation. These findings revealed the regulation mechanism of light-induced anthocyanin synthesis and provide excellent gene resources for breeding high-quality Tartary buckwheat.

## 1. Introduction

Tartary buckwheat (*Fagopyrum tataricum*), a model for both medicine and food, belongs to the Polygonaceae family. Besides starch, proteins, and other nutrients, Tartary buckwheat seeds are also rich in flavonoid content, especially rutin and quercetin, which are reported to be beneficial for human health due to their antioxidant, antidiabetic, anticancer, and anti-inflammatory traits, among others [1,2,3]. In addition, buckwheat seedlings have been reported to be rich in flavonoids, not only rutin and quercetin, but also anthocyanins, which are usually absent in seeds [4]. Nowadays, buckwheat seedlings have become prevalent as a kind of green and healthy vegetable in many countries. Tartary buckwheat is mainly cultivated in Asian countries, such as China, India, and Nepal, among others, and a small amount in European countries, such as Germany and Belgium [5]. It is reported that Eastern Tibet or Northwestern Yunnan of China is the center of origin of Tartary buckwheat [6], and China is the largest producer of Tartary buckwheat, where the cultivation area is about 400,000~600,000 hm^2^, accounting for about 80% plantation worldwide, and the annual output is between 300,000 and 500,000 tons [7]. The cotyledon and hypocotyl of buckwheat seedlings are red in color, which results from anthocyanin accumulation [8]. The anthocyanin biosynthesis pathway is one section of the flavonoid pathway, and the structural genes in the anthocyanin pathway include *chalcone synthase* (*CHS*), *chalcone isomerase* (*CHI*), *flavanone-3-hydroxylase* (*F3H*), *dihydroflavonol 4-reductase* (*DFR*), *anthocyanidin synthase* (*ANS*), and *UDP-glucose: flavonoid 3-glucosyltransferase* (*UFGT*), whose expression levels are mainly regulated by a protein complex, MBW [9,10]. This protein complex comprises of an MYB, a basic-helix-loop-helix (bHLH), and a WD40, and among these three types of transcript factors, MYB plays the main role in anthocyanin accumulation. In general, anthocyanin biosynthesis is induced by light, and this mechanism has been well studied in several plant species, such as *Arabidopsis* [11,12], pear [13,14], and apple [15,16], among others. In addition, high light intensity can promote anthocyanin accumulation [17,18]. Previous studies indicated that blue light, red light, far-red light, and UV facilitated anthocyanin biosynthesis [19,20,21,22]. Light-induced anthocyanin accumulation in plant tissue is a part of photomorphogenesis, and BBX (B-Box), the zinc finger transcription factor family, plays a vital role in this process through effecting light-signaling factors CONSTITUTIVE PHOTOMORPHOGENIC1 (COP1) and ELONGATED HYPOCOTYL5 (HY5) [11,12]. Previous studies on *Arabidopsis* demonstrated that different members of the BBX family played different roles in regulating photomorphogenesis under illumination and dark treatment. For example, AtBBX4, AtBBX20, AtBBX21, and AtBBX22 interact with COP1 to promote seedling light morphogenesis in the light [23,24], while in the dark, AtBBX24 and AtBBX25 interact with COP1, and are degraded by undergoing COP1-mediated ubiquitination. Additionally, they repress HY5 expression to prevent photomorphogenesis [25], otherwise, AtBBX24 may form an inactive heterodimer with HY5 that inhibits its transcription, resulting in negative regulation of light signaling [26]. Furthermore, AtBBX20, AtBBX22, and AtBBX23 can interact with HY5 and coordinately regulate the expression level of light-induced/-repressed genes, including genes in the anthocyanin pathway [11,23,27]. In apple, MdBBX20 and MdBBX22 are both capable to interacting with MdHY5 to enhance MdHY5 binding to its target genes *MdMYB1*, *MdMYB10*, *MdCHS*, *MdDFR*, and *MdANS*, which results in UV-B-induced anthocyanin accumulation in the peel of apple [15,16]. Meanwhile, it is suggested that MdHY5 might play a core role in regulating anthocyanin biosynthesis mediated by the BBXs–HY5 interaction since MdBBX24 and MdBBX33 can also interact with HY5 [16]. In pear pericarp, PpBBX16-PpHY5 positively regulates light-induced anthocyanin biosynthesis by activating *PpMYB10* [13]. Meanwhile, the other two BBX proteins, PpBBX18 and PpBBX21, competitively associate with PpHY5 to regulate anthocyanin biosynthesis [14]. When PpBBX18 interacted with PpHY5, the latter bound to the G-box motif of *PpMYB10* and the former provided the *trans*-acting activity to promote *PpMYB10* expression, resulting in anthocyanin accumulation. However, when PpBBx21 interacted with PpBBX18-PpHY5, it prevented the formation of the PpBBX18-PpHY5 complex, thus inhibiting anthocyanin pigmentation [14].

Although the mechanism of light-induced anthocyanin biosynthesis has been discovered in several plant species, this mechanism remains elusive in Tartary buckwheat. In this study, transcriptome analysis of Tartary buckwheat seedlings under light and dark treatments was undertaken in order to screen out some light-induced anthocyanin biosynthesis-related genes, and further verify the functions of these candidate genes by transgenic experiments and preliminarily explore the regulation mechanism of anthocyanin biosynthesis induced by light.

## 2. Results

### 2.1. Accumulation of the Total Anthocyanin Content of Tartary Buckwheat Sprouts

Tartary buckwheat sprouts growing in darkness had colorless hypocotyl with yellow cotyledon, while those growing under light treatment had red hypocotyl with green cotyledon (Figure 1A). The accumulations of anthocyanin in Tartary buckwheat sprouts under both treatments were measured, with results showing that sprouts in darkness contained almost no anthocyanin, while sprouts with illumination treatment had higher anthocyanin contents (Figure 1B).

### 2.2. Transcriptomic Analysis of Tartary Buckwheat Seedlings under Light and Dark Treatments through RNA-Seq

To obtain an in-depth understanding of the process of pigmentation induced by light, it is necessary to discover the expression profiles of genes involved in the biosynthesis and regulation of anthocyanin. cDNA libraries from one-week springs of Tartary buckwheat ‘Jinqiao 2’ under dark and light treatments were used to obtain the transcriptome profiles during their pigmentation process. Total read pairs generated from RNA-seq ranged from 19.3 to 23.0 million, and clean reads of each sample were 5.8–6.9 billion bp. High-quality 150 bp reads (Q > 20 and Q > 30) accounted for more than 97% and 92% of each clean reads, respectively, with GC content around 47%. Among them, 87.26–89.61% could be mapped on the ‘Pinku 1’ reference genome, with 82.74–86.89% and 2.09–5.09 mapping to the unique reads and multiple reads, respectively (Appendix A).

### 2.3. Analysis of Differentially Expressed Genes (DEGs)

The predicted total genes of ‘Pinku 1’ was 35,862; however, the expressed genes of these samples ranged from 24,776 to 25,380 (Appendix A). The values of FPKM (fragments per kilobase per million bases), representing gene expression levels of sprouts under different treatments, were used for analysis, and transcripts with FPKM values = 0 were the most abundant group (about 30%), which indicated that about one-third of the genes were not expressed. Here, 15~60 and 3~15 were the second most abundant groups, accounting for about 22% and 21%, respectively. Transcripts with FPKM values of more than 60 accounted for about 10% (Appendix A), and from 0~1 and 1~3, both less than 10% (Appendix A).

A total of 2774 genes were differently expressed between sprouts under dark and light treatments, with 1666 upregulated genes and 1110 downregulated genes in dark-treatment sprouts (Appendix A, Figure 2A). A hierarchical clustering map was obtained based on the gene expression patterns (Figure 2B), and light significantly affected the expression levels of most genes: some genes showed low expression levels and some expressed higher levels in both dark-treatment sprouts and light-treatment sprouts, and abundant genes exhibited higher expression profiles in the sprouts under light treatment (Figure 2B). Meanwhile, 12 DEGs were chosen to perform qRT-PCR to verify the reliability of the RNA-seq data, and high consistency between these data were obtained (Appendix A).

### 2.4. Functional Analysis of DEGs

Differentially expressed genes from the D vs. L group were subjected to GO and KEGG analysis to understand the functions of these DEGs and their possible contribution to the coloration process affected by light. Three categories, including the biology process, cellular component, and molecular function, were obtained, and each category contained 10–20 GO terms (Figure 3A). Among them, the biology process contained the largest number of terms (21), and DEGs related to the metabolic process, cellular process, and single-organism process were the most abundant. There were 16 terms contained in the cellular component, and DEGs involved in the cell, cell part, membrane, membrane part, and organelle occupied the majority. The category of molecular function only contained 10 terms, and catalytic activity and binding enriched more abundant DEGs (Figure 3A). In total, 20 significant pathways were identified by KEGG analysis (Figure 3B). Among which, carbon metabolism, phenylpropanoid biosynthesis, and starch and sucrose metabolism enriched the most abundant DEGs, followed by glycolysis/gluconeogenesis and cysteine and methionine metabolism (Figure 3B). Four pathways showed more significant enrichment, including photosynthesis-antenna proteins, porphyrin and chlorophyll metabolism, and carotenoid biosynthesis (Figure 3B), and this indicated that these four pathways were all associated with light.

### 2.5. DEGs Involved in the Anthocyanin Biosynthesis Pathway

KEGG enrichment analyses showed that the biosynthesis of secondary metabolites was the enriched pathway. Among them, 12 DEGs were related to the flavonoid pathway and most of them were more abundant in light-treated sprouts, and these key enzyme-encoding genes included 2 *cinnamate-4-hydroxylase* (*C4H*), 4 *chalcone synthase* (*CHS*), 2 flavanone-3-hydroxylase (*F3H*), 1 *flavonol synthase* (*FLS*), 1 *flavanone-4-reductase* (*DFR*), 1 *Anthocyanidin 3-O-glucosyltransferase* (*UFGT*), and 1 *anthocyanidin reductase* (*ANR*), as well as 1 *glutathione S-transferase* (*GST*), which is related to the transportation of flavonoids (Appendix A). *CHS*, *DFR*, *F3H*, *UFGT*, and *GST* led to anthocyanin biosynthesis and transport pathways, and except for one *CHS* gene, other anthocyanin-related genes were upregulated in sprouts grown in light (Figure 4).

Moreover, some transcription factors may be involved in the regulation of anthocyanin biosynthesis of Tartary buckwheat sprouts, which was induced by light. For example, one B-box protein gene (*FtPinG0009261700.01*), the homologous gene of *Arabidopsis AtBBX22*, named *FtBBX22*, and one *HY5* gene (*FtPinG0002935500.01*), a light-signaling factor, as well as one *R2R3*-*MYB* gene (*FtPinG0003544400.01*), which was annotated as anthocyanin regulatory C1 protein gene, and homologous with *ZmC1* of maize (Appendix A), it was named as *FtMYB42* and phylogenetically close to *AtMYB123/AtTT2* in our previous research [28]. AtBBX22 and AtHY5 have been identified to regulate photomorphogenesis [12,24], and they and ZmC1, as well as AtMYB123, were all reported to be involved in the positive regulation of anthocyanin biosynthesis in response to light, and there were interaction and regulation relationships among AtBBX22, AtHY5, and MYB [11,29,30]. Therefore, *FtBBX22*, *FtHY5*, and *FtMYB42*, which were all expressed higher in light-treatment Tartary buckwheat sprouts, may also have a similar function and interaction in the regulation of light-induced anthocyanin biosynthesis.

### 2.6. Subcellular Localization Analysis and Overexpression of FtBBX22, FtHY5, and FtMYB42 in Tobacco

To check the subcellular localization of FtBBX22, FtHY5, and FtMYB42, we conducted the transient expression signal of green fluorescent protein (GFP) fused with FtBBX22, FtHY5, and FtMYB42, respectively, and red fluorescent protein (RFP) as the nucleus location reference in *Nicotiana benthamiana* leaf epidermal cells. The results revealed that GFP-FtBBX22, GFP-FtHY5 and GFP-FtMYB42 all colocalized with RFP, respectively (Figure 5), indicating that all three proteins localized in the nucleus, and may play roles as transcription factors.

The functions of these three transcription factors were verified by overexpression in K326 tobacco, and compared with WT plant flowers, transgenic plant flowers had a darker red color (Figure 6A) and greater anthocyanin accumulations (Figure 6B). Most of the expression levels of anthocyanin biosynthesis-related genes, including *CHS*, *CHI*, *DFR*, *ANS*, and *UFGT*, were highly regulated in transgenic lines (*35S: FtBBX22*, *35S: FtHY5*, and *35S: FtMYB42*). Specifically, the *FtDFR* gene was expressed significantly higher in *FtBBX22*-overexpression transgenic lines by about 90-fold (Figure 6C), which was consistent with the total anthocyanin contents. These results suggested that all three transcription factors could promote anthocyanin accumulation by upregulating the expression level of structural genes in the anthocyanin biosynthesis pathway.

### 2.7. Trans-Acting Activity of FtBBX22 and FtHY5 on the Promoter of Anthocyanin Biosynthesis Genes FtDFR and FtMYB42, as Well as FtMYB42 on the Promoter of FtDFR

To understand how FtBBX22 induced anthocyanin biosynthesis, the correlations between FtBBX22 and anthocyanin biosynthesis-related genes (structural gene *FtDFR* and regulatory gene *FtMYB42*) were analyzed by the yeast one-hybrid assay, and the results suggested that FtBBX22 could not directly bind to the promoter regions of either *FtDFR* (Figure 7A) or *FtMYB42* (Figure 7B). Similar to FtBBX22, FtHY5 could not directly bind to the promoter regions of either *FtDFR* (Figure 7C) or *FtMYB42* (Figure 7D) yet. However, FtMYB42 could bind to the promoter of *FtDFR* (Figure 7E), suggesting that FtMYB42 may directly activate the expression of *FtDFR*, resulting in anthocyanin accumulation. Since the expression levels of most anthocyanin biosynthesis genes could be induced by FtBBX22 and FtHY5 in transgenic lines (Figure 6C), therefore, there may be some partner that FtBBX22 and FtHY5 combined with to indirectly regulate structural genes’ expression in the anthocyanin biosynthesis pathway.

### 2.8. The Interaction between FtBBX22 and FtHY5 and Their Joint Regulation on FtMYB42 and FtDFR Expression

HY5 acted as a partner of BBX22 in *Arabidopsis* [12] and BBX16 (a homolog of *Arabidopsis* BBX22) in pear [14] to promote photomorphogenesis. The physical interactions between FtBBX22 and FtHY5 were further analyzed by the BiFC assay and yeast two-hybrid assay. The results showed that FtBBX22 could bind to FtHY5 (Figure 8A,B); then, their association enhanced the expression level of *FtMYB42* and *FtDFR* by binding to their promoters, which was identified using the dual-luciferase assay (Figure 8C,D).

## 3. Discussion

Consumers have gained more and more interest in Tartary buckwheat sprouts due to their high content of flavonoids (especially rutin and anthocyanins), which are beneficial for human health, and low content of allergic proteins [31,32]. Since the availability of Tartary buckwheat genome information, numerous –omic studies focusing on different aspects of Tartary buckwheat have been conducted. Here, we first performed a comparative transcriptome analysis of Tartary buckwheat seedlings under light and dark treatments to screen anthocyanin biosynthesis-related genes and several regulators involved in light-induced anthocyanin accumulation. This was followed by the identification of some of their functions using biochemical assays.

Among the 2774 genes that showed differential expression between sprouts under dark and light treatments identified by RNA-seq (Figure 2A), DEGs involved in the flavonoid biosynthetic pathway were one significantly enriched group (Figure 3B), including 12 DEGs that were related to the flavonoid pathway (2 *C4H*, 4 *CHS*, 1 *FLS*, 1 *DFR*, 2 *F3H*, 1 *UFGT*, and 1 *ANR*; Appendix A). Of these, *CHS*, *DFR*, *F3H*, and *UFGT* catalyze the reactions in the anthocyanin branch of the flavonoid biosynthesis pathway, while *GST* is involved in the transportation of anthocyanins (Figure 4).

Except for these structural genes of the anthocyanin biosynthesis pathway, three transcription factor genes, including *FtBBX22*, *FtHY5*, and *FtMYB42*, were presumed to be involved in the regulation of light-induced anthocyanin accumulation. AtBBX22, a BBX22 family member, has been identified in photomorphogenesis, including anthocyanin pigmentation [23,24]. Meanwhile, AtBBX22 homologs in apple, MdCOL11 [33], and in red pear, PpBBX16 [13], were both reported to be involved in the regulation of light-induced anthocyanin biosynthesis. Moreover, our previous research showed that some members of the *BBX* family in Tartary buckwheat might regulate anthocyanin biosynthesis induced by light [32]. Here, a BBX member, *FtBBX22*, which was homologous with *AtBBX22*, exhibited a higher expression level in light-treatment Tartary buckwheat sprouts than that in sprouts under darkness and was very likely involved in the regulation of anthocyanin accumulation induced by light. The *FtHY5′* homolog is *AtHY5*, and the homologs of *FtMYB42* are *ZmC1* and *AtMYB123*. Since *AtHY5*, *ZmC1*, and *AtMYB123* were reported to be involved in the regulation of the anthocyanin biosynthesis process [11,24,29,30], therefore, *FtHY5* and *FtMYB42* may also have similar functions, and these hypotheses were verified by further analysis.

Subcellular location analysis showed that *FtBBX22*, *FtHY5*, and *FtMYB42* were located in the nucleus (Figure 5), indicating that they may act as transcription factors in the regulation of anthocyanin biosynthesis. Furthermore, their functions were verified by overexpression in K326 tobaccos, which have light-red flowers. The red colors of transgenic lines of flowers of these three genes all became darker (Figure 6A), and total anthocyanins contents of all three transgenic lines of flowers were higher than that of the WT plant flowers (Figure 6B). Additionally, the expression levels of most key genes in the anthocyanin biosynthesis pathway of these transgenic lines were significantly increased, especially in the *FtBBX22*-overexpression transgenic tobacco flowers (Figure 6C), which were consistent with the phenotypes (Figure 6A) and anthocyanin accumulations (Figure 6B). All these results suggested that these three transcription factors positively regulated anthocyanin accumulation by upregulating some of the anthocyanin biosynthesis-related genes, and these results were similar to those in apple [15,16,34] and red pear [13,14,35].

Although all the anthocyanin biosynthesis-related genes were significantly upregulated in *FtBBX22/FtHY5/FtMYB42*-overexpression transgenic tobacco flowers, especially *NtDFR* in *FtBBX22*-overexpression transgenic lines (Figure 6C), FtBBX22 and FtHY5 could not directly bind to the promoter of *FtDFR* or *FtMYB42* in the yeast one-hybrid assays (Figure 7). However, *FtMYB42* could directly bind to the promoter of *FtDFR* (Figure 7E), which is similar to MYBs in most plants [36]. This suggested that there may be other proteins that interact with FtBBX22, and together regulate anthocyanin-related genes. Our further analysis indicated that FtHY5 could physically interact with FtBBX22, which was identified by BiFC and yeast two-hybrid assays (Figure 8A,B), and dual-luciferase assays showed that the FtBBX22/FtHY5 protein complex activated the expression of *FtMYB42* and *FtDFR* (Figure 8C,D). Therefore, FtBBX22 might need the partner FtHY5 to play the role in the regulation of light-induced anthocyanin accumulation. These associations among BBX, HY5, MYB, and anthocyanin biosynthesis genes in Tartary buckwheat were consistent with those in red pear [13] and apple [15,16]. However, how they change in the Tartary buckwheat sprouts growing in darkness, in which anthocyanin accumulation was repressed, needs further research.

## 4. Materials and Methods

### 4.1. Plant Materials

The seeds of Tartary buckwheat (‘Jinqiao 2’) were obtained from the School of Life Sciences, Research Center of Buckwheat Industry Technology, Guizhou Normal University. Germination was carried out according to the paper-bed germination method [32]. Briefly, after sterilization by 2% NaClO and washing, uniform size seeds were laid on 10 cm × 10 cm sprouting boxes, whose bottom was covered with filter paper, ddH_2_O was added, and they were put in illumination incubators for germination. The light condition was set as 16 h light/8 h dark, 10,000 Lx, 80% humidity, and 25 °C, and the dark condition was set as 24 h dark, 80% humidity, and 25 °C. Irrigation with ddH_2_O was conducted every two days. After germination for 7 days, seedlings cultivated under light and dark treatments were collected, respectively, and frozen in liquid N_2_ immediately and stored at −80 °C for further analysis. Three biological repeats for each sample were performed.

### 4.2. RNA Extraction and RNA-Seq

Totals RNA of the samples mentioned above was extracted using the GREENspin plus plant RNA kit (Zomanbio, Beijing, China) according to the manufacturer’s manual, and three biological replicates were performed for each sample. The genomic DNA was removed by RNase-free DNase I (Takara, Dalian, China). RNA qualities were preliminarily evaluated by running 1% agarose gel, and then the purity and concentration were measured using the NanoPhotometer^®^ spectrophotometer (IMPLEN, Calabasas, CA, USA). The value of the RNA integrity number (RIN) was assessed using the RNA Nano 6000 Assay Kit of the Bioanalyzer 2100 system (Agilent Technologies, Santa Clara, CA, USA).

In total, 3 μg of RNA of each sample with RIN ≥ 7 was used to construct the Illumina sequencing library according to the recommendations of the NEBNext^®^ Ultra^TM^ RNA Library Prep Kit for Illumina^®^ (NEB, Ipswich, MA, USA). These 6 libraries were sequenced by the Illumina Hiseq 2000 at Frasergen Company (Wuhan, China) and 150 bp paired-end reads were used to sequence.

### 4.3. Identification and Analysis of Differentially Expressed Genes (DEGs)

Clean reads were obtained after removing those reads that contained adapters, with the percentage of poly-N ≥10% of this read, and low-quality reads with the number of bases (quality value SQ ≤ 5) taking up over 50% of this read. The contents of Q20, Q30, and GC of clean data were assessed, then clean reads were aligned to the reference genome sequence of ‘Pinku 1’ [37], which can be downloaded from (https://www.ncbi.nlm.nih.gov/genome/38383, accessed on 13 November 2023), for gene annotation. The value of fragments per kilobase per million fragments (FPKM) was used to present the gene expression level.

Differentially expressed genes (DEGs) were detected using the DESeq2 of R package (1.10.1) with |Log_2_ (fold-change)| > 1 and false discovery rate (FDR) < 0.05.

Analysis of Gene Ontology (GO) and Kyoto Encyclopedia of Genes and Genomes (KEGG) of DEGs were carried out. GO terms with FDR < 0.05 were chosen as significantly enriched and classified into three major groups, including biological process, cellular component, and molecular function. Pathways obtained by KEGG analysis with FDR < 0.05 were described as significantly enriched pathways to better understand the function of these DEGs.

### 4.4. RNA-Seq Validation by Quantitative Real-Time PCR (qRT-PCR)

About 1 μg of total RNAs from each sample were synthesized into cDNAs using a First-Strand cDNA Synthesis Kit (TOYOBO, Osaka, Japan) according to the manufacturer’s instructions. Some differentially expressed genes were chosen to perform qRT-PCR, and the primer sequences are listed in Appendix A. Here, 20 μL of qRT-PCR reaction system from iQ^TM^ SYBR Green Super mix (Bio-Rad, Santa Clara, CA, USA) were carried out on the C1000^TM^ thermal cycler coupled with a CFX96^TM^ Detection Module (Bio-Rad, Berkeley, USA). The program was as follows: 95 °C for 3 min, then 40 cycles at 95 °C for 10 s, 60 °C for 30 s, and 72 °C for 10 s. Each sample corresponded to that of the RNA-seq experiment and was performed in three replicates. The Tartary buckwheat *Actin* gene (*FtPinG0002124000.01*) [32,38] and tobacco *NtActin* [39] were used as the reference genes, and the relative expression levels of these genes were calculated with the 2^−ΔΔCt^ method.

### 4.5. Measurement of Total Anthocyanin Content

Total anthocyanin content was analyzed by following the previous method [28,32]. In brief, 1 g of sample was powered using liquid nitrogen, then mixed well with 4 mL of methanol containing 1% (*v*/*v*) HCl and incubated for 24 h at 4 °C. After centrifugation at 12,000× *g* rpm for 10 min, the supernatant of the mixture was collected, and the absorbance value was measured at 530 nm and 657 nm, respectively. The anthocyanin content was calculated by the formula: Q_anthocyanin_ = (A_530_ − 0.25 × A_657_) × M^−1^ (where M is the fresh weight of the sample). Three biological repeats for each sample were analyzed

### 4.6. Subcellular Localization Analysis

The subcellular locations of FtBBX22, FtHY5, and FtMYB42 were detected following the previous method [40]. Briefly, the CDS of FtBBX22, FtHY5, and FtMYB42 were amplified and inserted into the pC1300-GFP vector, respectively (primers are listed in Appendix A), in which fusion genes were driven by the 35S promoter. *Agrobacteria* of 35S: GFP-FtBBX22, 35S: GFP-FtHY5, and 35S: GFP-FtMYB42 were, respectively, co-infiltrated with H2H-RFP (the nucleus marker) into 3-week-old *Nicotiana benthamiana* leaves. Signals were observed under a confocal microscope (Zeiss 980 Confocal Microscope, Leica, Wetzlar, Germany) at 36 to 48 h after infiltration.

### 4.7. Transgenic Analysis of Red-Flower Tobacco

The CDS sequences of *FtBBX22*, *FtHY5,* and *FtMYB42* were obtained by PCR using high-fidelity thermostable DNA polymerase (Vazyme, China) and specific primers (listed in Appendix A), and then ligated into the overexpression vector pC1300 using the homologous recombination method, respectively. These recombinant plasmids were transformed into tobacco (K326, with red flower) using the leaf disc method [41]. The positive transgenic tobacco plants were used for further study.

The total anthocyanin content of transgenic and wild-type tobacco flowers was detected using the method described in Section 4.5.

The primers of genes involved in anthocyanin biosynthesis in tobacco for qRT-PCR analysis are listed in Appendix A, and *NtActin* [39] was used as the reference gene. The qRT-PCR program was performed as mentioned in Section 2.4.

### 4.8. Yeast One-Hybrid Assay

Promoter fragments of *FtDFR* and *FtMYB42* were ligated into the pHIS II vector, and genes (*FtBBX22*, *FtHY5*, and *FtMYB42*) were cloned into the pGADT7 vector. The primers of these genes are listed in Appendix A. Five pairs: pHIS II-promoter *FtDFR* and pGADT7- *FtBBX22*, pHIS II-promoter *FtMYB42 and* pGADT7- *FtBBX22*, pHIS II-promoter *FtDFR* and pGADT7-*FtHY5*, pHIS II-promoter *FtMYB42* and pGADT7-*FtHY5*, and pHIS II-promoter *FtDFR* and pGADT7-*FtMYB42*, were simultaneously transformed into Y109 yeast strains, respectively, then tested on SD/-Leu/-Trp plates, and further tested on SD/-His/-Leu/-Trp + 3AT plates. Meanwhile, the pair pGADT7-p53 and pHIS II-p53 was used as the positive control, and the pairs pGADT7-p53 and pHIS II-promoter *FtDFR/FtMYB42* and pGADT7-*FtBBX22*/*FtHY5/FtMYB42* and pHIS II-p53 were used as negative controls, respectively.

### 4.9. Bimolecular Fluorescence Complementation (BiFC) Assay

The CDSs of *FtBBX22* and *FtHY5* without the termination codon were ligated into the YFP^C^ (p2YC) and YFP^N^ (p2YN; primers listed in Appendix A), respectively, and transformed into *Agrobacteria* GV3101, respectively, and then co-transformed into the onion epidermis. Fluorescence was observed after 24 h of cultivation on MS medium via a Zeiss 980 Confocal Microscope (Leica, Wetzlar, Germany).

### 4.10. Yeast Two-Hybrid Assay

The yeast two-hybrid assay was carried out using the Matchmaker Gold Yeast Two-Hybrid System Kit (Takara, Dalian, China), in accordance with the manufacturer’s instructions. Briefly, 500~600 ng of pGADT7-*FtBBX22* vector and pGBKT7-*FtHY5* vector was simultaneously transformed into Y2H Gold yeast cells, and the transformants were tested on SD/-Leu/-Trp plates, and further selected on Trp/-Leu/-His/-Ade/X-a-gal plates. The pGADT7-T and pGBKT7-53, and pGADT7-T and pGBKT7-Lam were used as positive and negative controls, respectively. The primers of these genes are listed in Appendix A.

### 4.11. Dual-Luciferase Assay

Dual-luciferase assays were performed using *N. benthamiana* according to the previous method described by Niu et al. [42]. In brief, the full-length CDS of *FtBBX* and *FtHY5* genes were cloned into the pGreenII 0029 62-SK vector, respectively, while the promoter sequences of *FtMYB42* and *FtDFR* were inserted into the pGreenII 0800-LUC vector, respectively. Both constructs were individually transformed into *Agrobacterium* GV3101 (containing the pSoup vector). Then, 72 h after infiltration, the values of the activities of Firefly luciferase and Renilla luciferase were analyzed by the microplate reader (VICTOR Nivo^TM^, PerkinElmer, Waltham, MA, USA). Both luciferase activities were analyzed with three biological replications for each assay.

## 5. Conclusions

Here, comparative transcriptome analysis of Tartary buckwheat sprouts under light and dark treatments was performed to identify DEGs involved in anthocyanin biosynthesis, including structural genes and transcription factor genes. The three TFs included a BBX family member gene, *FtBBX22*, a *FtHY5* gene, and a *MYB* gene, *FtMYB42*, all promoting anthocyanin accumulation, and further analysis showed that FtBBX22 interacted with HY5, then this complex activated the expression of *FtMYB42* and *FtDFR*, leading to light-induced anthocyanin biosynthesis. Our findings provided the BBX22-HY5-MYB42 regulatory module involved in light-induced anthocyanin biosynthesis in Tartary buckwheat.

## Figures and Tables

**Figure 1 ijms-25-08376-f001:**
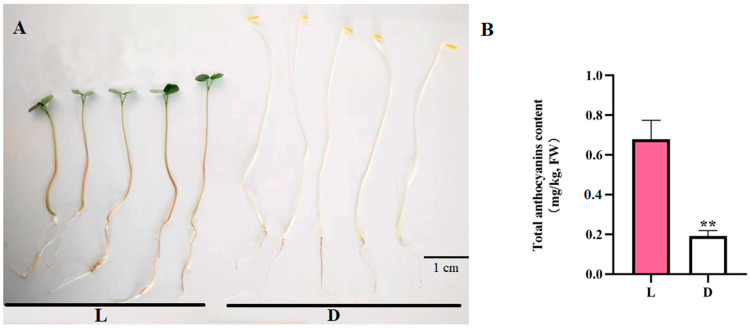
(**A**) Tartary buckwheat ‘Jiniqiao 2’ sprouts under light (L) and dark (D) treatments, (**B**) and their total anthocyanin contents (** *p* < 0.01).

**Figure 2 ijms-25-08376-f002:**
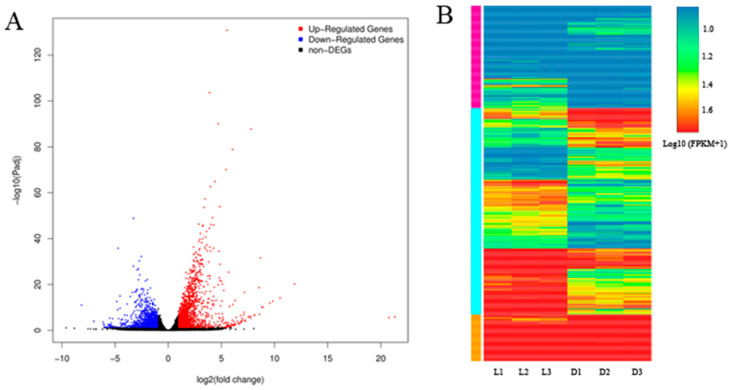
Profiles of differentially expressed genes. (**A**) Volcano plots of differentially expressed genes, with upregulated, downregulated, and no change in DEGs. (**B**) Hierarchical clustering map of DEGs.

**Figure 3 ijms-25-08376-f003:**
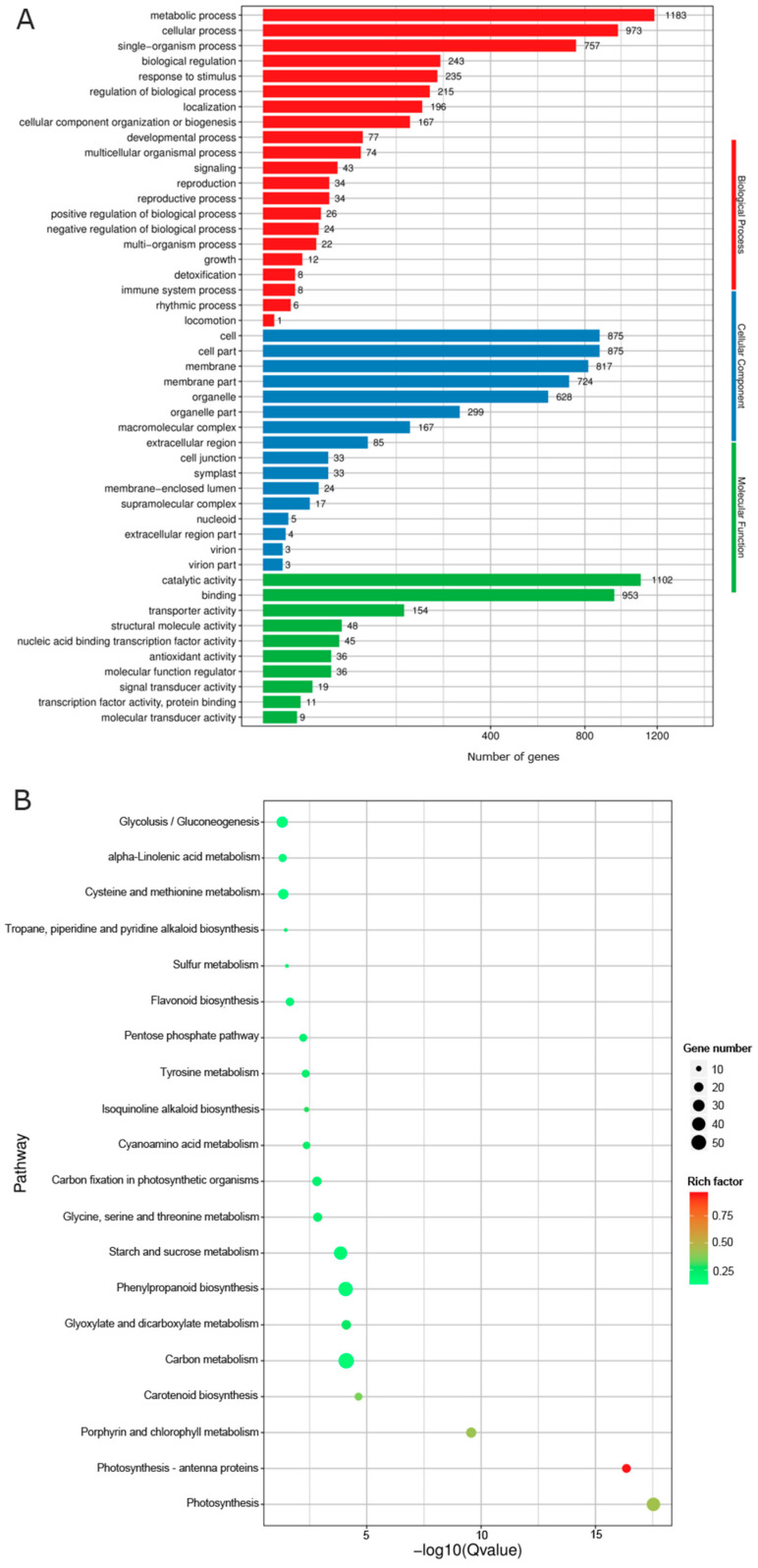
Functional analysis of DEGs. (**A**) GO analysis of DEGs and (**B**) KEGG analysis of DEGs.

**Figure 4 ijms-25-08376-f004:**
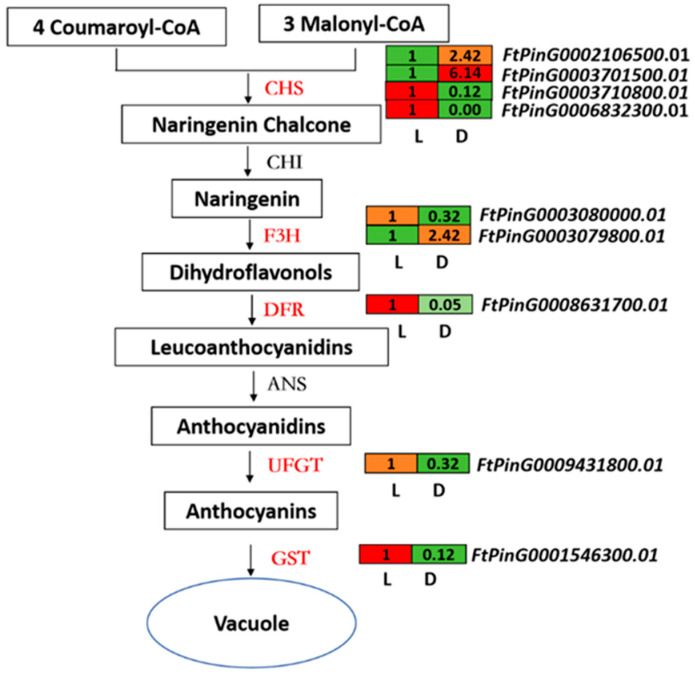
DEGs related to structural genes of anthocyanin biosynthesis and transportation pathways. Enzymes in red and green font indicate that they were up- and down-regulated between L (light treatment) and D (dark treatment) sprouts, respectively, and the digitized heatmap beside the changed enzyme-encoding genes shows their relative expression levels.

**Figure 5 ijms-25-08376-f005:**
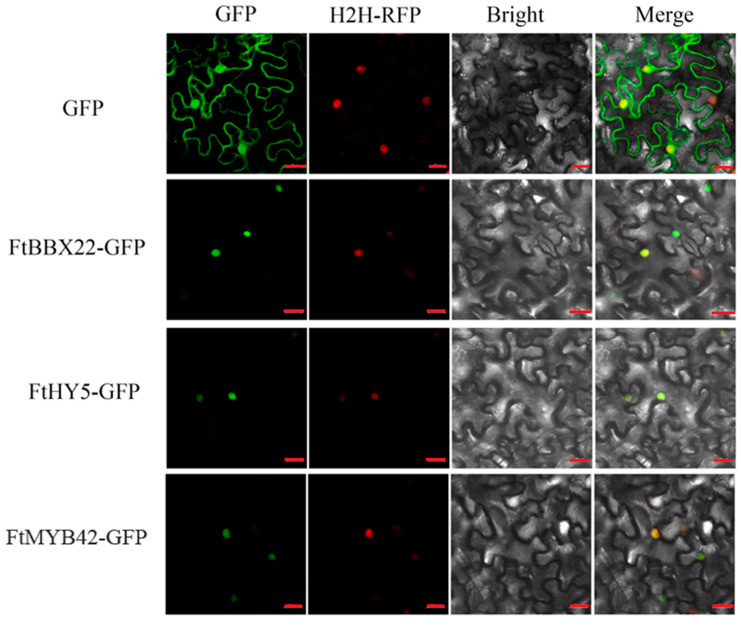
Subcellular localization of FtBBX22, FtHY5, and FtMYB42 in *Nicotiana benthamiana* leaf epidermal cells. GFP, GFP fluorescence; H2H-RFP, nucleus marker; Merge, merge of GFP, RFP, and bright-field images. Bar = 20 μm.

**Figure 6 ijms-25-08376-f006:**
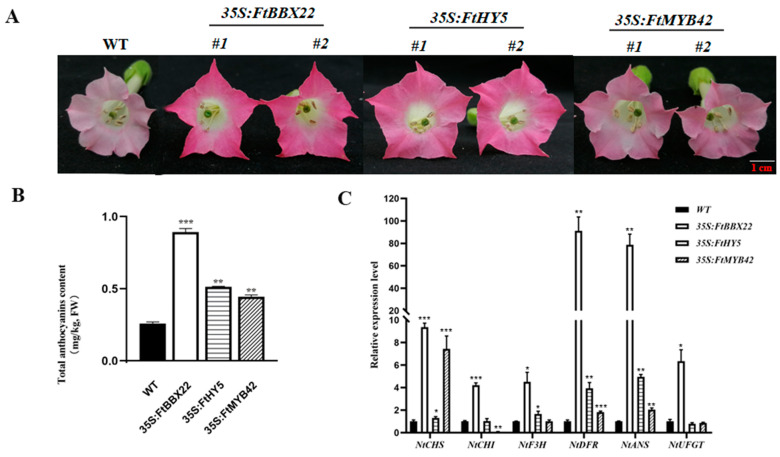
Effects of *FtBBX22*, *FtHY5*, and *FtMYB42* overexpression in K326 tobacco. (**A**) Flowers of wild K326 tobacco and transgenic tobacco of overexpression of *FtBBX22*, *FtHY5*, and *FtMYB42*. (**B**) The total anthocyanin contents in the transgenic tobacco. (**C**) The expression levels of anthocyanin biosynthesis genes in transgenic lines (significant analyses were performed between each kind of transgenic line (*35S: FtBBX22/35S: FtHY5*/*35S: FtMYB42*) and WT plant by TTEST, respectively, * *p* < 0.05, ** *p* < 0.01, and *** *p* < 0.001).

**Figure 7 ijms-25-08376-f007:**
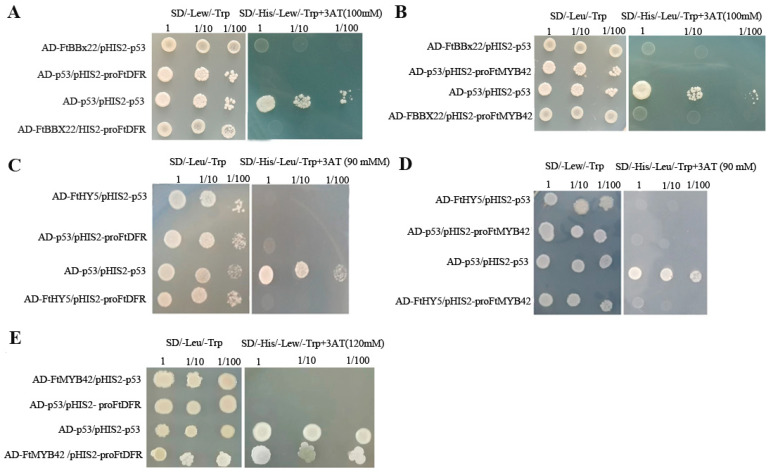
(**A**) Yeast one-hybrid assay of FtBBX22 and the promoter of *FtDFR*. (**B**) Yeast one-hybrid assay of FtBBX22 and the promoter of *FtMYB42*. (**C**) Yeast one-hybrid assay of FtHY5 and the promoter of *FtDFR*. (**D**) Yeast one-hybrid assay of FtHY5 and the promoter of *FtMYB42*. (**E**) Yeast one-hybrid assay of FtMYB42 and the promoter of *FtDFR*.

**Figure 8 ijms-25-08376-f008:**
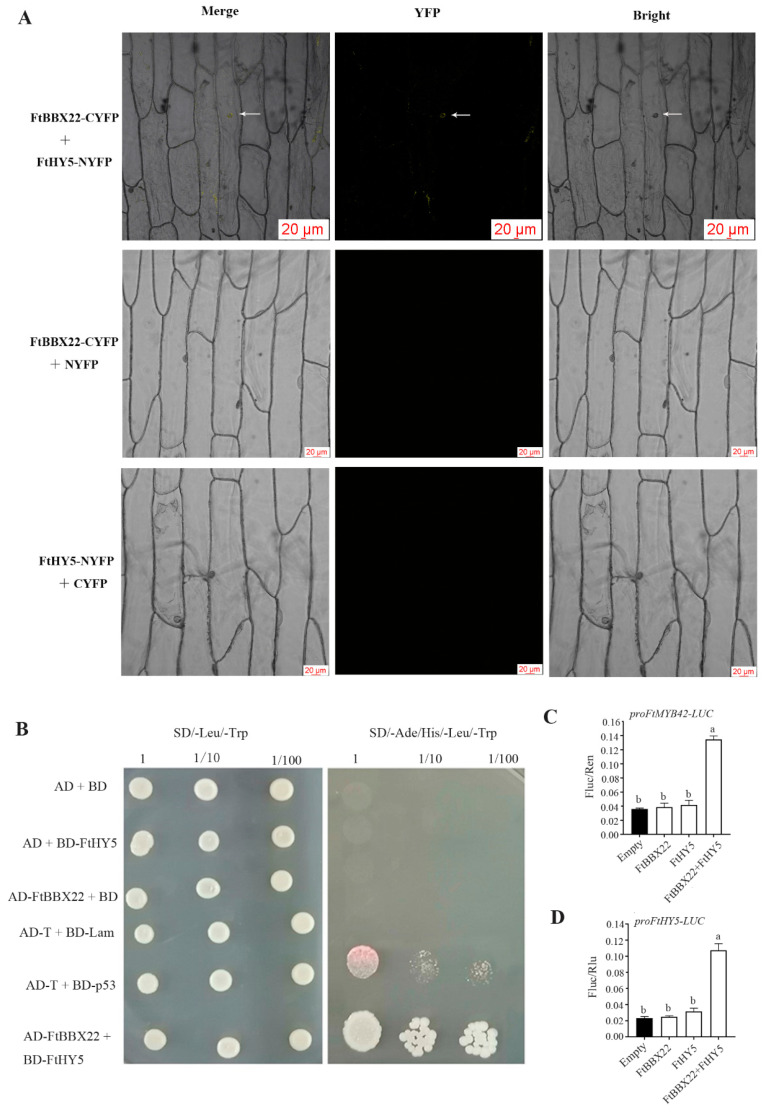
The interaction between FtBBX22, FtHY5, and FtMYB42 as well as anthocyanin biosynthesis gene *FtDFR*. The physical interaction of FtBBX22 and FtHY5 was tested by BiFC assays (arrows indicate the nucleus) (**A**) and yeast two-hybrid assays (**B**). FtBBX22 and FtHY5 jointly promoted the expression of *FtMYB42* (**C**) and *FtDFR* (**D**). Lowercase letters above bars indicate a significant difference determined by two-way ANOVA followed by multiple comparisons with Tukey’s test (*p* < 0.05).

## Data Availability

Data are contained within the article and Appendix A.

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
