# Peer review of "The Complex FtBBX22 and FtHY5 Positively Regulates Light-Induced Anthocyanin Accumulation by Activating FtMYB42 in Tartary Buckwheat Sprouts"

_ijms, 2024, doi:10.3390/ijms25158376_

Round 1
Reviewer 1 Report
Comments and Suggestions for Authors
The authors of the manuscript “The complex FtBBX2 and FtHY5 positively regulates light-induced anthocyanins accumulation by activating FtMYB42 in Tartary buckwheat sprouts” accomplished transcriptome analysis of Tartary buckwheat (Fagopyrum tataricum ‘Jiniqiao 2’) seedlings grown under light and dark treatments to identify the roles of one B-box protein BBX22 and HY5 in anthocyanin synthesis. They also performed a subcellular localization, transgenic and yeast one-hybrid analysis. The results showed that cooperation between FtBBX22 and FtHY5 leads to the co-activation of the transcription of FtMYB42, resulting in the accumulation of anthocyanin through the binding to the FtDFR promoter. This study can help enhance the nutritional content of Tartary buckwheat. The manuscript is well-written, and the results are discussed.
Specific comments:
Abstract:
L13: Tartary buckwheat (Fagopyrum tataricum).
L14: requires light ‘induction’. Please delete ‘induction’.
Introduction:
L36: Please indicate the countries and annual production of Tartary buckwheat.
L46: In general, anthocyanin synthesis occurs or is enhanced under stress conditions such as high light intensities and UV stress. Please indicate light quality and intensity's role in anthocyanin synthesis for better understanding because the seedlings were grown under 10000 lux.
Results:
L95: Please improve the quality of Figure 1A.
L96: Please indicate the statistical significance (Figure 1B **).
L149: Please improve the quality of Figure 3B.
L227: Please improve the quality of Figure 7.
Discussion:
Please avoid subheadings.
Materials and Methods:
L309: Please indicate the source of the seeds, pot/tray size, potting mix, and irrigation/fertigation details.
L312: 10 000 Lx, delete it.
Comments on the Quality of English Language
Minor editing of English language required
Author Response
|
Response to Reviewer 1 Comments
|
||
|
|
|
|
|
1. Comments and Suggestions for Authors The authors of the manuscript “The complex FtBBX2 and FtHY5 positively regulates light-induced anthocyanins accumulation by activating FtMYB42 in Tartary buckwheat sprouts” accomplished transcriptome analysis of Tartary buckwheat (Fagopyrum tataricum ‘Jiniqiao 2’) seedlings grown under light and dark treatments to identify the roles of one B-box protein BBX22 and HY5 in anthocyanin synthesis. They also performed a subcellular localization, transgenic and yeast one-hybrid analysis. The results showed that cooperation between FtBBX22 and FtHY5 leads to the co-activation of the transcription of FtMYB42, resulting in the accumulation of anthocyanin through the binding to the FtDFR promoter. This study can help enhance the nutritional content of Tartary buckwheat. The manuscript is well-written, and the results are discussed.
Summary: Thank you very much for taking the time to review this manuscript. Please find the detailed responses below and the corresponding revisions/corrections highlighted/in track changes in the re-submitted files.
|
||
|
2. Questions for General Evaluation |
Reviewer’s Evaluation |
Response and Revisions |
|
Does the introduction provide sufficient background and include all relevant references? |
Can be improved |
We have improved it |
|
Is the research design appropriate? |
Yes |
|
|
Are the methods adequately described? |
Can be improved |
We have improved it |
|
Are the results clearly presented? |
Can be improved |
We have improved it |
|
Are the conclusions supported by the results? |
Yes |
|
|
3. Point-by-point response to Comments and Suggestions |
||
|
Comments 1: Abstract: L13: Tartary buckwheat (Fagopyrum tataricum). Response 1: Thank you for pointing this out, we have add (Fagopyrum tataricum) after Tartary buckwheat. Comments 2: Abstract: L14: requires light ‘induction’. Please delete ‘induction’. Response 2: Thank you for your suggestion, we have deleted ‘induction’.
Comments 3: Introduction: L36: Please indicate the countries and annual production of Tartary buckwheat. Response 3: Thank you for pointing this out, we have add this information in the text as follows: Tartary buckwheat is mainly cultivated in Asia countries, such as China, India, Nepal among others., and a small amount in European countries, such as Germany and Belgium [5]. It’s reported that the Eastern Tibet or North-western Yunnan of China is the center of origin of Tartary buckwheat [6], and China is the largest producer of Tartary buckwheat where cultivation area of Tartary buckwheat is about 400,000 ~600,000 hm², accounting for about 80% plantation worldwide, and the annual output is between 300,000 and 500,000 tons [7]. 5. Luthar Z, Golob A, Germ M, Vombergar B, Kreft I. (2021).Tartary buckwheat in human nutrition. Plants (Basel). 10, 700. doi: 10.3390/plants10040700. 6. Konishi T, Yasui Y, Ohnishi O. (2005). Original birthplace of cultivated common buckwheat inferred from genetic relationships among cultivated populations and natural populations of wild common buckwheat revealed by AFLP analysis. Genes. Genet. Syst. 80, 113-9. doi: 10.1266/ggs.80.113. 7. Dong, X. N., Tang, Y., Ding, M. Q., Zhu, L., Li, J. B., Wu, Y. M., Shao, J. R. (2017). Germplasm resources of buckwheat in China and their forage value. Pratacul. Sci. 34, 378-388. (in Chinese)
Comments 4: Introduction: L46: In general, anthocyanin synthesis occurs or is enhanced under stress conditions such as high light intensities and UV stress. Please indicate light quality and intensity's role in anthocyanin synthesis for better understanding because the seedlings were grown under 10000 lux. Response 4: Thank you for pointing this out. We have added the content of light quality and intensity's role in anthocyanin synthesis in the text as follows: In addition, high light intensity can promote anthocyanins accumulation [17,18]. Previous studies indicated that blue light, red light, far-red light, and UV facilitated anthocyanins biosynthesis [19-22]. 17. Zhang, Y., Xu, S., Cheng, Y., Peng, Z., Han, J. (2018). Transcriptome profiling of anthocyanin-related genes reveals effects of light intensity on anthocyanin biosynthesis in red leaf lettuce. PeerJ. 6, e4607. doi: 10.7717/peerj.4607. 18. Li, J., He, Y. J., Zhou, L., Jiang, M., Liu, Y., Zhou, L., He, Y. et al. (2018). Transcriptome profiling of genes related to light-induced anthocyanin biosynthesis in eggplant(Solanum melongena L.)before purple color becomes evident. BMC Genomics. 19, 201. doi: 10.1186/s12864-018-4587-z. 19. Liu, Z., Zhang, Y., Wang, J., Li, P., Zhao, C., Chen, Y., Bi, Y. (2015). Phytochrome-interacting factors PIF4 and PIF5 negatively regulate anthocyanin biosynthesis under red light in Arabidopsis seedlings. Plant Sci. 238, 64-72. doi: 10.1016/j.plantsci.2015.06.001. 20. Fu, Z., Shang, H., Jiang, H., Gao, J., Dong, X., Wang, H., Zhang, H. (2020). Systematic identification of the light-quality responding anthocyanin synthesis-related transcripts in petunia petals. Horticul. Plant J. 6, 428–438. doi:10.1016/j.hpj.2020.11.006. 21. Van Brenk, J. B., Courbier, S., Kleijweg, C. L., Verdonk, J. C., Marcelis, L. F. M. (2024) Paradise by the far-red light: Far-red and red:blue ratios independently affect yield, pigments, and carbohydrate production in lettuce, Lactuca sativa. Front. Plant Sci. 15,1383100. doi: 10.3389/fpls.2024.1383100. 22. Li, W., Tan, L., Zou, Y., Tan, X., Huang, J., Chen, W., Tang, Q. (2020). The effects of ultraviolet A/B Treatments on anthocyanin accumulation and gene expression in dark-purple tea cultivar 'Ziyan' (Camellia sinensis). Molecules. 25, 354. doi: 10.3390/molecules25020354.
Comments 5: Results: L95: Please improve the quality of Figure 1A. Response 5: Thank you for pointing this out. We have improved the quality of Figure1A. Comments 6: Results: L96: Please indicate the statistical significance (Figure 1B **). Response 6: Thank you for pointing it, we have indicate the statistical significance (Figure 1B **) in the legend of Figure 1.
Comments 7: Results: L149: Please improve the quality of Figure 3B. Response 7: We have improved the quality of Figure 3B.
Comments 8: Results: L227: Please improve the quality of Figure 7. Response 8: We have improved the quality of Figure 7.
Comments 9: Discussion: Please avoid subheadings. Response 9: Thank you for your suggestion, we have avoided subheadings.
Comments 10: Materials and Methods: L309: Please indicate the source of the seeds, pot/tray size, potting mix, and irrigation/fertigation details. Response 10: Thank you for pointing this out. We have added this information in the text as follows: The seeds of Tartary buckwheat (‘Jinqiao 2’) were obtained from school of Life Sciences, Research Center of Buckwheat Industry Technology, Guizhou Normal University. Germination was carried out according to the paper bed germination method [32]. Briefly, after sterilization by 2% NaClO and washing, uniform size seeds were laid on 10 cm ×10 cm sprouting boxes, whose bottom was covered with filter paper, ddH2O was added and put in illumination incubators for germination. Light condition was set as 16 h light/8h dark, 10, 000 Lx, 80% humidity and 25 oC, and dark condition was set as 24 h dark, 80% humidity and 25 oC. Irrigation with ddH2O were conducted every two days. After germination for 7 days, seedlings cultivated under light and dark treatment were collected, respectively, and frozen in liquid N2 immediately and stored at -80 oC for further analysis. Three biological repeats for each sample were performed.
Comments 11: Materials and Methods: Materials and Methods: L312: 10 000 Lx, delete it Response 11: Thank you for pointing this out, we have deleted “10 000 Lx” |
||
|
4. Response to Comments on the Quality of English Language |
||
|
Minor editing of English language required Response : we have improved the quality of English in this paper by an expert in scientific communication in molecular studies, the details are as follows: (1) Line 18-19, “Overexpression assay showed that FtHY5 and FtBBX22 both can promote anthocyanin synthesis…” has been changed to “Overexpression assay showed that FtHY5 and FtBBX22 both could promote anthocyanin synthesis…”. |
||
|
(2) Line 20, “…with FtHY5 to form a complex that activate the transcription of a MYB transcription…” has been changed to “…with FtHY5 to form a complex that activates the transcription of a MYB transcription…”. |
||
|
(3) Line 31-32, “…due to antioxidant, antidiabetic, anticancer, and anti-inflammatory…” has been changed to “due to their antioxidant, antidiabetic, anticancer, and anti-inflammatory traits…”.
(4) Line 37, ” The cotyledon and hypocotyl of buckwheat seedlings exhibit red color, “ has been changed to ” The cotyledon and hypocotyl of buckwheat seedlings are red in color, “.
(5) Line 43-44, “This protein complex is comprised of an MYB, a basic-helix-loop- helix (bHLH) and a WD40,” has been changed to “This protein complex comprises of an MYB, a basic-helix-loop- helix (bHLH) and a WD40,”.
(6) Line 54, “…family played different roles in the regulation of photomorphogenesis under illumination…” has been changed to “…family played different roles in regulating of photomorphogenesis under illumination…”.
(7) Line 57, “AtBBX24 and AtBBX25 interact with COP1, and degraded by…” has been changed to “AtBBX24 and AtBBX25 interact with COP1, and is degraded by…”
(8) Line 60, “…inactive heterodimer with HY5 to inhibit its transcription,” has been changed to “…inactive heterodimer with HY5 that inhibits its transcription,”
(9) Line 64, “In apple, MdBBX20 and MdBBX22 both can interact with…” has been changed to ” n apple, MdBBX20 and MdBBX22 are both capable to interacting with…“
(10) Line 66, “which resulted in UV-B-induced anthocyanin accumulation…” has been changed to “which results in UV-B-induced anthocyanin accumulation…”
(11) Line 80-81, “…transcript analysis of Tartary buckwheat seedlings under light and dark treatments were undertaken.” has been changed to “…transcriptome analysis of Tartary buckwheat seedlings under light and dark treatments was undertaken.”
(12) Line 89, “Tartary buckwheat sprouts growing in darkness were colorless in the hypocotyl…” has been changed to “Tartary buckwheat sprouts growing in darkness had colorless hypocotyl…”
(13) Line 92, “which showed that sprouts in darkness…” has been changed to “with results showing that sprouts in darkness…”
(14) Line 93, “however, sprouts with illumination treatment had…” has been changed to “while sprouts with illumination treatment had…”
(15) Line 101, “it is necessary to discover the expression profiles of genes that involved in the biosynthesis…” has been changed to “it is necessary to discover the expression profiles of genes involved in the biosynthesis…”
(16) Line 111, “The predicted total genes of ‘Pinku 1’ are 35,862, ” has been changed to “The predicted total genes of ‘Pinku 1’ is 35,862, ”
(17) Line 115, “which indicated that about one-third genes didn’t express.” has been changed to “which indicated that about one-third genes were not expressed.”
|
||
|
(18) Line 133, “Differentially expressed genes from D VS L group were subjected to GO and KEGG…” has been changed to “Differentially expressed genes from the D VS L group were subjected to GO and KEGG…”
(19) Line 142, “Totally, 20 significant pathway…” has been changed to “In total, 20 significant pathways…”
(20) Line 147, “…porphryrin and chlorophyll metabolism and carotenoid biosynthesis…” has been changed to “…porphyrin and chlorophyll metabolism and carotenoid biosynthesis…”
(21) Line 152, “KEGG enrichment analyses showed that biosynthesis of secondary metabolites…” has been changed to “KEGG enrichment analyses showed that the biosynthesis of secondary metabolites…”
(22) Line 153, “Among them, 12 DEGs were related to flavonoid pathway…” has been changed to |
||
|
“Among them, 12 DEGs were related to the flavonoid pathway…”
(23) Line 160-161, “other anthocyanin-related genes were up regulated in sprouts growing in light…” has been changed to “other anthocyanin-related genes were up regulated in sprouts grown in light…” |
||
|
|
||
|
(24) Line 163, “Figure 4. DEGs that related structural genes of anthocyanin biosynthesis…”has been changed to “Figure 4. DEGs related to structural genes of anthocyanin biosynthesis” |
||
(25) Line 179-181, “which all expressed higher in light-treatment Trartary 179 buckwheat sprouts, and may also have the similar function and interaction in regulation 180 of light-induced anthocyanin biosynthesis.” has been changed to “which all expressed higher in light-treatment Tartary buckwheat sprouts, may also have a similar function and interaction in the regulation of light-induced anthocyanin biosynthesis.”
(26) Line 184-184, “ FtBBX22, FtHY5 and FtMYB42” has been changed to “ FtBBX22, FtHY5, and FtMYB42”
(27) Line 189-190, “and may play roles of transcript factor.” has been changed to “and may play roles of transcription factor.”
(28) Line 200, “especially DFR gene expressed significantly higher…” has been changed to “Specifically, the FtDFR gene was expressed significantly higher…”
(29) Line 202, “These results suggested that all these three transcription factors…” has been changed to “These results suggested that all three transcription factors…”
(30) Line 216, “…FtMYB42 (Fig. 7B), while FtHY5 couldn’t directly bind to the promoter regions of…” has been changed to “FtMYB42 (Fig. 7B). Similar to FtBBX22, FtHY5 couldn’t directly bind to the promoter regions of…”
(31) Line 218-219, “suggesting that FtMYB42 may directly activate the expression of FtDFR.” has been changed to “suggesting that FtMYB42 may directly activate the expression of FtDFR, resulting in anthocyanins accumulation.”
(32) Line 219-220, “The expression levels of most of anthocyanin biosynthesis genes could be 219 induced by FtBBX22 in transgenic lines s (Fig. 6C),” has been changed to “Since the expression levels of most anthocyanin biosynthesis genes could be induced by FtBBX22 and FtHY5 in transgenic lines (Fig. 6C), therefore, there may be some partner that FtBBX22 and FtHY5 combined with to indirectly regulate structural genes expression in anthocyanin biosynthesis pathway.”
(33) Line 246-247, “This was followed by identification of some of their functions using biochemical assays.” has been changed to “This was followed by the identification of some of their functions using biochemical assays.”
(34) Line 264, “…previous research showed that some members of BBX family in Tartary buckwheat might…” has been changed to “…previous research showed that some members of the BBX family in Tartary buckwheat might…”
(35) Line 286, “…by up regulating some of anthocyanin biosynthesis-related genes,” has been changed to “…by up-regulating some of anthocyanin biosynthesis-related genes,”
(36) Line 291-292, “Although all the anthocyanin biosynthesis-related genes were significantly up regulated…” has been changed to “Although all the anthocyanin biosynthesis-related genes were significantly up-regulated…”
(37) Line 295, “But FtMYB42 could directly bind to the promoter of FtDFR (Fig .7E),” has been changed to “However, FtMYB42 could directly bind to the promoter of FtDFR (Fig .7E)”
(38) Line 301-301, “Therefore, FtBBX22 might need the partner FtHY5 in playing the role on the regulation of light-induced anthocyanin accumulation.” has been changed to “Therefore, FtBBX22 might need the partner FtHY5 to play the role in the regulation of light-induced anthocyanin accumulation.”
(39) Line 318-320, “RNAs quality were preliminarily 318 evaluated by running 1% agarose gel; and then the purity and concentration was measured using…” has been changed to “RNA qualities were preliminarily evaluated by running 1% agarose gel; and then the purity and concentration were measured using…”
(40) Line 330, “…and low quality reads with the number of bases…” has been changed to “and low- quality reads with the number of bases…”
(41) Line 332, “…clean data were assessment,” has been changed to “clean data were assessed,”
(42) Line 349, “…primer sequence are listed in Table S1.” has been changed to “…primer sequences are listed in Table S1.”
(43) Line 353, “Each sample was corresponded to that for RNA-Seq experiment…” has been changed to “Each sample was corresponded to that of the RNA-Seq experiment…”
(44) Line 360, “and inserted into pC1300-GFP vector,” has been changed to “and inserted into the pC1300-GFP vector,”
(45) Line 374, “Total anthocyanin content of transgenic and wild-type tobacco flowers was detected 373 using previously described method (Zhao et al., 2021; Deng et al., 2021). “ has been changed to ” Total anthocyanin content of transgenic and wild-type tobacco flowers was detected using the method described in 4.5 section.“
Reviewer 2 Report
Comments and Suggestions for Authors
The manuscript describes a detailed study to identify genes involved in light-induced anthocyanins accumulation and their interaction. The experiments are well designed though some information must be added. Also, at least an explicit objective must be included and some formal corrections should be done for having concordance between Material and Methods and Results. I have made the respective comments in the attached file.

A native English with expertise in scientific communication in molecular studies should revise the manuscript.
Author Response
|
Response to Reviewer 2 Comments
|
|||||||||
|
|
|
|
|||||||
|
1. Comments and Suggestions for Authors The manuscript describes a detailed study to identify genes involved in light-induced anthocyanins accumulation and their interaction. The experiments are well designed though some information must be added. Also, at least an explicit objective must be included and some formal corrections should be done for having concordance between Material and Methods and Results. I have made the respective comments in the attached file.
Summary: Thank you very much for taking the time to review this manuscript. Please find the detailed responses below and the corresponding revisions/corrections highlighted/in track changes in the re-submitted files.
|
|||||||||
|
2. Questions for General Evaluation |
Reviewer’s Evaluation |
Response and Revisions |
|||||||
|
Does the introduction provide sufficient background and include all relevant references? |
Yes |
|
|||||||
|
Is the research design appropriate? |
Can be improved |
|
|||||||
|
Are the methods adequately described? |
Must be improved |
We have improved it |
|||||||
|
Are the results clearly presented? |
Must be improved |
We have improved it |
|||||||
|
Are the conclusions supported by the results? |
Can be improved |
|
|||||||
|
3. Point-by-point response to Comments and Suggestions |
|||||||||
|
Comments 1: The final paragraph of Introduction is expected to present the objectives of the research, which is not this case. Here, a presentation of results and conclusions is preliminarily done. Response 1: Thank you for pointing this out, we have changed it as follows: Although the mechanism of light-induced anthocyanin biosynthesis has been discovered in several plant species, this mechanism remains elusive in Tartary buckwheat. In this study, transcriptome analysis of Tartary buckwheat seedlings under light and dark treatments were undertaken in order to screen out some light-induced anthocyanin biosynthesis-related genes, and further verify the functions of these candidate genes by transgenic experiments and preliminarily explore the regulation mechanism of anthocyanin biosynthesis induced by light. Comments 2: This experiment is not presented in Material and Methods. Please, add a description including the statistical analyses. Response 2: Thank you for pointing this out. We have added this this experiment in Material and Methods as follows: 4.5 Measurement of total anthocyanins content Total anthocyanin content were analyzed by following the previous method [28,32]. In brief, 1 g sample was powered using liquid nitrogen, then mixed well with 4 mL methanol containing 1% (v/v) HCl and incubated for 24 h at 4 oC. After centrifugation at 12,000 rpm for 10 min. The supernatant of mixture was collected, and the absorbance value was measured at 530 nm and 657 nm, respectively. The anthocyanin content was calculated by the formula Qanthocyanin = (A530 − 0.25 × A657) × M−1 (M is the fresh weight of the sample). Three biological repeats for each sample were analyzed Additionally, we have indicate the statistical significance in the legend of Figure 1.
Comments 3: 2.7 section: In Material and Methods, these experiments are independently described. Please, rewrite the subsections in a unique one, as presented here, including the statistical analyses. Response 3: Thank you for this suggestion, we have rewrote the subsection in a unique one as follows: 2.7. Trans-acting activity of FtBBX22, FtHY5 on the promoter of anthocyanin biosynthesis genes FtDFR and FtMYB42, as well as FtMYB42 on the promoter of FtDFR To understand how FtBBX22 and FtHY5 induced anthocyanin biosynthesis, the correlations between FtBBX22/FtHY5 and anthocyanin biosynthesis-related genes (structural gene FtDFR and regulatory gene FtMYB42) were analyzed by yeast one-hybrid assays, and the results suggested that FtBBX22 could not directly bind to the promoter regions of either FtDFR (Fig. 7A) or FtMYB42 (Fig. 7B). Similar to FtBBX22, FtHY5 couldn’t directly bind to the promoter regions of either FtDFR (Fig. 7C) or FtMYB42 (Fig. 7D) yet. However, FtMYB42 could bind to the promoter of FtDFR (Fig. 7E), suggesting that FtMYB42 may directly activate the expression of FtDFR, resulting in anthocyanins accumulation. Since the expression levels of most anthocyanin biosynthesis genes could be induced by FtBBX22 and FtHY5 in transgenic lines (Fig. 6C), therefore, there may be some partner that FtBBX22 and FtHY5 combined with to indirectly regulate structural genes expression in anthocyanin biosynthesis pathway.
2.8. The interaction between FtBBX22 and FtHY5 and they together regulation on FtMYB42 and FtDFR expression HY5 acted as a partner of BBX22 in Arabidopsis [12] and BBX16 (a homolog of Arabidopsis BBX22) in pear [14] to promote photomorphogenesis. The physical interaction between FtBBX22 and FtHY5 were further analyzed by BiFC assay and yeast two-hybrid assay, the results showed that FtBBX22 can bind to FtHY5 (Fig. 8A and 8B), then their association enhanced to the expression level of FtMYB42 and FtDFR by binding to their promoters, which was identified using dual-luciferase assay (Fig. 8C and 8D). Additionally, the statistical analysis method was indicated in the legends of Figure 6 and figure 8.
Comments 4: 10000 Lx in 24 h dark. Please, revise this phrase. Response 4: Thank you for pointing this out. We have deleted “10000 Lx” and provide more detail of plant materials as follows: The seeds of Tartary buckwheat (‘Jinqiao 2’) were from school of Life Sciences, Research Center of Buckwheat Industry Technology, Guizhou Normal University. Germination was carried out according to the paper bed germination method [32]. Briefly, after sterilization by 2% NaClO and washing, uniform size seeds were layed on 10 cm ×10 cm sprouting boxes, whose bottom was covered with filter paper, then proper ddH2O added and put in illumination incubators for germination. Light condition was set as 16 h light/8h dark, 10, 000 Lx, 80% humidity and 25 oC, and dark condition was set as 24 h dark, 80% humidity and 25 oC. Irrigation with proper ddH2O were conducted every two days. After germination for 7 days, seedlings cultivated under light and dark treatment were collected, respectively, and frozen in liquid N2 immediately and stored at -80 oC for further analysis. Three biological repeats for each sample were performed.
|
|||||||||
|
4. Response to Comments on the Quality of English Language |
|||||||||
|
A native English with expertise in scientific communication in molecular studies should revise the manuscript.
(25) Line 179-181, “which all expressed higher in light-treatment Trartary 179 buckwheat sprouts, and may also have the similar function and interaction in regulation 180 of light-induced anthocyanin biosynthesis.” has been changed to “which all expressed higher in light-treatment Tartary buckwheat sprouts, may also have a similar function and interaction in the regulation of light-induced anthocyanin biosynthesis.”
(26) Line 184-184, “ FtBBX22, FtHY5 and FtMYB42” has been changed to “ FtBBX22, FtHY5, and FtMYB42”
(27) Line 189-190, “and may play roles of transcript factor.” has been changed to “and may play roles of transcription factor.”
(28) Line 200, “especially DFR gene expressed significantly higher…” has been changed to “Specifically, the FtDFR gene was expressed significantly higher…”
(29) Line 202, “These results suggested that all these three transcription factors…” has been changed to “These results suggested that all three transcription factors…”
(30) Line 216, “…FtMYB42 (Fig. 7B), while FtHY5 couldn’t directly bind to the promoter regions of…” has been changed to “FtMYB42 (Fig. 7B). Similar to FtBBX22, FtHY5 couldn’t directly bind to the promoter regions of…”
(31) Line 218-219, “suggesting that FtMYB42 may directly activate the expression of FtDFR.” has been changed to “suggesting that FtMYB42 may directly activate the expression of FtDFR, resulting in anthocyanins accumulation.”
(32) Line 219-220, “The expression levels of most of anthocyanin biosynthesis genes could be 219 induced by FtBBX22 in transgenic lines s (Fig. 6C),” has been changed to “Since the expression levels of most anthocyanin biosynthesis genes could be induced by FtBBX22 and FtHY5 in transgenic lines (Fig. 6C), therefore, there may be some partner that FtBBX22 and FtHY5 combined with to indirectly regulate structural genes expression in anthocyanin biosynthesis pathway.”
(33) Line 246-247, “This was followed by identification of some of their functions using biochemical assays.” has been changed to “This was followed by the identification of some of their functions using biochemical assays.”
(34) Line 264, “…previous research showed that some members of BBX family in Tartary buckwheat might…” has been changed to “…previous research showed that some members of the BBX family in Tartary buckwheat might…”
(35) Line 286, “…by up regulating some of anthocyanin biosynthesis-related genes,” has been changed to “…by up-regulating some of anthocyanin biosynthesis-related genes,”
(36) Line 291-292, “Although all the anthocyanin biosynthesis-related genes were significantly up regulated…” has been changed to “Although all the anthocyanin biosynthesis-related genes were significantly up-regulated…”
(37) Line 295, “But FtMYB42 could directly bind to the promoter of FtDFR (Fig .7E),” has been changed to “However, FtMYB42 could directly bind to the promoter of FtDFR (Fig .7E)”
(38) Line 301-301, “Therefore, FtBBX22 might need the partner FtHY5 in playing the role on the regulation of light-induced anthocyanin accumulation.” has been changed to “Therefore, FtBBX22 might need the partner FtHY5 to play the role in the regulation of light-induced anthocyanin accumulation.”
(39) Line 318-320, “RNAs quality were preliminarily 318 evaluated by running 1% agarose gel; and then the purity and concentration was measured using…” has been changed to “RNA qualities were preliminarily evaluated by running 1% agarose gel; and then the purity and concentration were measured using…”
(40) Line 330, “…and low quality reads with the number of bases…” has been changed to “and low- quality reads with the number of bases…” (41) Line 332, “…clean data were assessment,” has been changed to “clean data were assessed,”
(42) Line 349, “…primer sequence are listed in Table S1.” has been changed to “…primer sequences are listed in Table S1.”
(43) Line 353, “Each sample was corresponded to that for RNA-Seq experiment…” has been changed to “Each sample was corresponded to that of the RNA-Seq experiment…”
(44) Line 360, “and inserted into pC1300-GFP vector,” has been changed to “and inserted into the pC1300-GFP vector,”
(45) Line 374, “Total anthocyanin content of transgenic and wild-type tobacco flowers was detected 373 using previously described method (Zhao et al., 2021; Deng et al., 2021). “ has been changed to ” Total anthocyanin content of transgenic and wild-type tobacco flowers was detected using the method described in 4.5 section.“
|
|||||||||
|
|
|||||||||
|
|
|||||||||
|
|
|||||||||
|
|
|||||||||
|
|
|||||||||
|
|
|||||||||